# Transcription Factor ERF194 Modulates the Stress-Related Physiology to Enhance Drought Tolerance of Poplar

**DOI:** 10.3390/ijms24010788

**Published:** 2023-01-02

**Authors:** Xuhui Huan, Xingqi Wang, Shengqiang Zou, Kai Zhao, Youzhi Han, Shengji Wang

**Affiliations:** College of Forestry, Shanxi Agricultural University, Jinzhong 030801, China

**Keywords:** gene expression pattern, drought stress, physiological mechanism, genetic transformation *ERF194*

## Abstract

Drought is one of the main environmental factors limiting plant growth and development. The AP2/ERF transcription factor (TF) ERF194 play key roles in poplar growth and drought-stress tolerance. However, the physiological mechanism remains to be explored. In this study, the *ERF194*-overexpression (OX), suppressed-expression (RNA interference, RNAi), and non-transgenic (WT) poplar clone 717 were used to study the physiology role of ERF194 transcription factor in poplar growth and drought tolerance. Morphological and physiological methods were used to systematically analyze the growth status, antioxidant enzyme activity, malondialdehyde (MDA), soluble sugars, starch, and non-structural carbohydrate (NSC) contents of poplar. Results showed that, compared with WT, OX plants had decrease in plant height, internode length, and leaf area and increased number of fine roots under drought stress. In addition, OX had higher water potential, activities of superoxide dismutase (SOD), catalase (CAT) and peroxidase (POD), contents of chlorophyll, soluble sugar, starch, and NSC, implying that ERF194 positively regulates drought tolerance in poplar. The growth status of RNAi was similar to those of WT, but the relative water content and CAT activity of RNAi were lower than those of WT under drought treatment. Based on the transcriptome data, functional annotation and expression pattern analysis of differentially expressed genes were performed and further confirmed by RT-qPCR analysis. Gene ontology (GO) enrichment and gene expression pattern analysis indicated that overexpression of *ERF194* upregulated the expression of oxidoreductases and metabolism-related genes such as *POD* and *SOD*. Detection of *cis*-acting elements in the promoters suggested that ERF194 may bind to these genes through MeJA-responsive elements, ABA-responsive elements, or elements involved in defense and stress responses. The above results show that ERF194 improved tolerance to drought stress in poplar by regulating its growth and physiological factors. This study provides a new idea for the role of ERF194 transcription factor in plant growth and drought-stress response.

## 1. Introduction

Currently, drought is one of the major abiotic stressors affecting plant survival and development and has become a growing problem in many parts of the world [1]. Drought stress can cause a range of physiological and morphological changes in plants, including stomatal closure, inhibition of cell growth, accumulation of reactive oxygen species, limitation of photosynthesis, as well as affecting the composition of phytochromes [2,3,4]. Severe drought stress may cause growth stagnation and leaf wilting and even plant death [4,5]. To adapt to drought conditions, plants have evolved morphological and physiological balancing mechanisms. For example, plants enhance their resistance to drought stress by regulating stomatal opening, promoting fine root growth, increasing abscisic acid content and scavenging the accumulation of reactive oxygen species, and enhancing photosynthesis [6,7]. Research suggests that the frequency, duration, and intensity of droughts will increase significantly due to future global warming [8]. Moreover, extreme drought can kill millions of trees in a short time [9,10]. 

Poplar is one of the most adaptable trees and grows worldwide. Poplar can not only restore vegetation and soil salinization, but also play roles in biomass development and utilization of lignocellulose energy plants, which has great potential value. However, poplar has great demand for water resources [7]. The hybrid poplar clone 717 (*Populous tremolo* × *P. Alba*) has strong potential application value in vegetation restoration, soil erosion control, and saline–alkali restoration. It is also a high-quality lignocellulose energy factor in the development of biomass energy [11]. Therefore, it is necessary to conduct an in-depth study on the physiological mechanisms of poplar in response to drought stress.

Transcription factors (TFs) play an important role in responding to various stresses by regulating the expression of stress-related genes in a synchronous manner [12,13]. APETALA2/ethylene-response factor (AP2/ERF) family is an important member of the transcription factor families. According to the AP2 domain similarity, 145 AP2/ERF transcription factor genes in *Arabidopsis* were divided into five categories, namely AP2 (APETALA2), RAV (related to ABI3/VP1), SOLOIST, DREB (dehydration-responsive element-binding protein), and ERF (ethylene-responsive factor) [14]. The AP2 subfamily is specific to developmental events and characterized by two highly similar and consecutively repeated AP2 domains. The RAV subfamily consists of an AP2/ERF domain and a B3 domain. It plays an important role in ethylene response and biotic and abiotic stress response [15]. The SOLOIST contains an AP2 domain whose amino acid sequence and gene structure are quite different from other AP2/ERF transcription factors. The DREB and ERF subfamilies are mainly composed of a single AP2/ERF domain, and the main difference between them is the 14th and 19th amino acid residues of the AP2 domain [14,16]. Members of the AP2/ERF family are involved in a variety of plant biological processes, including plant growth and development, bacterial defense, and a variety of environmental stresses such as drought and salt stress [16,17]. ERF6 and ERF11 antagonize mannitol-induced growth inhibition in *Arabidopsis* [18]. *MicroRNA172* and its *APETALA2*-like target genes regulate flowering time and floral organ characteristics in *Arabidopsis* [19]. ERF76 can activate the expression of stress-related genes in transgenic poplar and improve the salt tolerance of transgenic plants [20]. PagERF16 promotes lateral root proliferation and sensitizes to salt stress in poplar [21]. AP2/ERF transcription factor GmDREB1 confers drought tolerance in transgenic soybean by interacting with GmERFs [14]. TaAP2-15 is positively involved in the resistance of wheat to stripe rust fungus [22]. NbERF-IX-33 is involved in the production of phytoalexins to enhance the resistance of *Nicotiana benthamiana* to *Phytophthora infestans* [23]. MdERF38 promotes apple anthocyanin biosynthesis under drought stress [6]. 

*ERF194* is homologous to *ERF016* (AT5G21960.1) of *Arabidopsis thaliana* and belongs to the DREB A-5 subgroup. DREB subfamily members can recognize drought- and cold-induced response elements (DRE/CRT, A/GCCGAC) and play a very important role in plant resistance to abiotic stress [24,25,26]. In previous studies, it was found that the expression of *ERF194* was induced by NaCl, KCl, CdCl_2_, and PEG treatments [27]. Our former studies have indicated that ERF194 played an important role in poplar growth and drought stress endurance [28,29]. In this study, based on our previous studies, we focused on the growth and physiological indicators of ERF194 transgenic poplar under drought stress to explore the physiological mechanisms of ERF194 transcription factors in the drought tolerance of poplar. This series of studies will systematically reveal the function of the ERF194 transcription factor in poplar drought-stress tolerance.

## 2. Results

### 2.1. ERF194 Enhanced Drought Tolerance through Regulating Aboveground Part

After treatment with drought stress, RNAi showed the worst growth status, followed by WT, and OX had the best growth status (Figure 1a). Then, we monitored the growth indices under normal and drought conditions. Under normal watering conditions (CK), both transgenic poplar and non-transgenic poplar continued to increase in plant height, and RNAi had the highest plant height, but OX had the shortest plant height (*p* = 0.027) (Figure 1b). When treated with drought stress (DR), the plant height of transgenic poplar and non-transgenic poplar increased slowly, and there was no significant difference among OX, WT, and RNAi. In addition, the plant height of all the tested lines under DR conditions was significantly lower than that under CK, indicating that drought stress significantly inhibited the growth of plant height (Figure 1b). The base diameter showed a similar trend to the plant height although the difference between control and treated groups was not significant (Figure 1c). The 4th internode length of OX was shorter than WT and RNAi, which may contribute to the semi-dwarf phenotype of OX (*p* = 0.001, *p* = 0.002) (Figure 1d). The number of leaves of OX, WT, and RNAi were also detected, but the increase of leaf number of OX did not have much positive effect on the semi-dwarf plant height. However, an increasing number of leaves can contribute to photosynthesis and increase photosynthetic products (*p* = 0.019, *p* = 0.000) (Figure 1e). The chlorophyll content, which is always positively related to stress-endurance ability of OX, was significantly higher than that of WT and RNAi under both drought and normal watering conditions (*p* = 0.003, *p* = 0.000) (Figure 1f). Leaf size is the result of a plant’s trade-off between heating and water, and plants have evolved small leaves to adapt to severe environments in hot and arid regions. In this study, the leaves of OX were significantly smaller than those of other two lines, which made a positive contribution to the drought tolerance of OX (*p* = 0.002, *p* = 0.001, *p* = 0.003, *p* = 0.002) (Figure 2).

### 2.2. ERF194 Enhanced Drought Tolerance through Promoting Fine Roots Growth

As we know, the inhibition of root growth under drought stress is mainly reflected in the inhibition of absorbing root growth [30]. The absorption function of roots is mainly performed by fine roots with a diameter less than 2 mm [30]. Under CK, the surface area, volume, and length of the total roots of RNAi were lower than those of the other two lines, while the root of OX was robust (*p* = 0.012, *p* = 0.05, *p* = 0.005) (Figure 3a–c). After DR, the surface area and length of the roots of each line were significantly reduced. However, the surface area, volume, and the root length of OX were 51.69%, 46.73%, and 56.98% bigger than that of WT, respectively (*p* = 0.005, *p* = 0.011, *p* = 0.002) (Figure 3a–c). Under both CK and DR conditions, the average diameter of roots of OX was lower than that of WT. However, the root diameter of RNAi was similar to WT under CK conditions but smaller when treated with DR (*p* = 0.001, *p* = 0.021) (Figure 3d). Under CK and DR conditions, the conducting root length of RNAi was higher than that of WT (*p* = 0.011) (Figure 3e), while the absorbing root length of OX was significantly higher than that of WT (*p* = 0.000) (Figure 3f). Under CK conditions, the specific root length and specific root surface of RNAi were significantly lower than that of WT. On the contrary, the root tissue density of RNAi was significantly higher than that of WT (*p* = 0.001, *p* = 0.003, *p* = 0.026), whereas there was no significant difference among the three lines under DR (Figure 3g–i). After treatment with drought stress, compared with WT, OX roots were longer, and absorbing roots were denser (Figure 3j). Specific root length, referring to the total root length of fine roots per unit mass, is an important index to characterize the morphology and physiological function of fine roots [31]. In a word, the largest absorbing and specific root length of OX indicate that ERF194 can positively regulate plant tolerance to drought stress through promoting fine roots growth.

### 2.3. ERF194 Improved Physiological Performance of Poplar under Drought Stress

Malonaldehydic acid (MDA) is one of membrane lipid peroxidation products [32]. The results showed that there was no significant difference in MDA content among different poplar lines under normal conditions. After DR, MDA content of OX was 8.74% lower than that of WT (*p* = 0.019) (Figure 4a). Superoxide dismutase (SOD) is the main protective enzyme of the membrane lipid peroxidation defense system. It can catalyze the disproportionation reaction of reactive oxygen species to generate non-toxic molecular oxygen, thereby preventing plants from being poisoned. Catalase (CAT) and peroxidase (POD), which can remove H_2_O_2_ in physiological systems, also play protective function [33,34]. Under CK, CAT activity of OX was lower than those of the other two lines (*p* = 0.002). However, there was no significant difference in POD and SOD activities among the three lines. After DR, the activities of CAT, POD, and SOD in OX were 44.44%, 32.02%, and 50.74% higher than those in WT, respectively. Compared with WT, RNAi had higher CAT activity but lower SOD activity (*p* = 0.000, *p* = 0.017, *p* = 0.000) (Figure 4b–d).

Plant water potential can directly reflect the degree of plant water deficit, and plants can adjust their water potential to alleviate water deficit under drought [35]. Leaf water potential is the most sensitive physiological indicator of plant water-deficit status [36]. Under CK, there was no significant difference in both water potential and relative water content among the three lines. After DR, the water potential and relative water content of each line decreased, while the water potential and relative water content of OX were 24.94% and 7.04 higher than WT, respectively. However, the relative water content of RNAi was 16.22% lower than that of WT (*p* = 0.022, *p* = 0.002) (Figure 4e,f).

Soluble sugar, one of the components of non-structural carbohydrates (NSC) in forest trees, has functions in osmotic regulation, signal transduction, and cavitation repair [37,38]. In general, water stress can cause an increase in soluble sugar content and a decrease in starch content in leaves [39]. Under CK, the contents of soluble sugar, starch, and NSC of OX were significantly lower than those of WT and RNAi (*p* = 0.009, *p* = 0.030). When treated with drought stress, the increased contents of soluble sugar, starch, and NSC of OX were higher than those of WT and RNAi (*p* = 0.001, *p* = 0.004) (Figure 4g–i).

### 2.4. ERF194 Induced the Expression of Stress-Related Genes

Based on transcriptome data of hybrid poplar clone 717, 10 stress-tolerance-related genes including *ERF194* were screened for GO annotation and enrichment analysis to identify and predict the molecular functions of differentially expressed genes and the biological processes they involved (Figure 5). According to the biological function, these genes were mainly divided into three categories: molecular function, cellular component, and biological process (Figure 5). Molecular functions are mainly manifested in antioxidant activity and binding and catalytic activity. Cellular component is mainly manifested in the cell part, and biological processes are mainly manifested as metabolic process, cellular process, and response to stimulus (Figure 5a). GO enrichment showed that these genes were enriched in a series of important GO terms such as antioxidant activity, response to stress, reactive oxygen species metabolic process, response to stimulus and oxidoreductase activity, and acting on peroxide as acceptor (Figure 5b).

Co-expression analysis results showed that *ERF194* was directly related to *Potri.001G070900* and *Potri.002G013200,* which were involved in oxidative stress, peroxidase activity, and abscisic acid response process (Figure 6j, Table 1). Based on the RNA-Seq data, nine genes related to stress endurance were selected to detect the molecular function of ERF194 in this study (Figure 6). RT-qPCR was used to verify the expression of these genes in responding to drought stress (Figure 6). Under normal conditions, the expression of these stress-related genes in WT is independent of the high or low expression of OX and RNAi, whereas gene expression in OX was significantly induced and higher than that in RNAi and WT plants under drought stress. Of note, the expression of *Potri.008G065600, Potri.005G195600* and *SOD2* were induced in OX but suppressed in RNAi compared to WT (*p* = 0.000, *p* = 0.001, *p* = 0.002) (Figure 6d,f,i). This indicated that overexpression of *ERF194* could up-regulate the expression of stress-related genes to enhance the drought tolerance of poplar. Through physiological experiments and transcriptome sequencing data, we found that ERF194 affected the activity of antioxidant enzymes and the expression of related genes. To verify whether the promoters of these antioxidant enzyme genes contain *cis*-acting elements to which ERF194 can bind, we predicted and determined the locations of *cis*-elements in the promoters of these genes using the PlantCRAE (Appendix A). Most of the promoters of these genes have multiple stress-related *cis*-acting elements, such as ABRE, G-Box, MBS, and ARE motifs [40,41]. ERF194 may bind to these genes through the MeJA response elements, ABA response elements, or elements involved in defense and stress responses. However, the interaction relationship needs further study.

## 3. Discussion

In this study, we found that ERF194 transcription factor enhances drought tolerance by regulating plant growth and physiology. The morphology and physiological metabolism of plants are the results of the interaction effect between gene expression and environmental factors [42]. Through analysis, it was found that drought has effects on the inhibiting of plant growth, but ERF194 can alleviate this inhibition. Through the analysis of plant height, basal diameter, and stem node length, we found that ERF194 had a regulatory trend of dwarfing and stem node increasing (Figure 1). Dwarf plants are generally characterized by low plant height, small crown size, easy management, and high resistance to stress [42]. Therefore, OX plants exhibited greater drought tolerance probably associated with shorter plant height. As the major organ for photosynthesis, the leaf not only undertakes the function of gas exchange between the plant and the environment but also regulates the water transport [43]. The overexpression of *ERF194* significantly changed the leaf shape of hybrid poplar clone 717, with shorter, narrower, and smaller girth and area of leaves (Figure 2). Plants in arid areas often reduce water loss by reducing leaf area to accommodate drought stress [44]. Changes in leaf morphology of OX made it more drought-tolerant than RNAi and WT. In addition, the chlorophyll content of OX leaves also increased significantly, and drought stress had little effect on its content (Figure 1), which greatly increased the photosynthesis rate and promoted the production of photosynthetic products such as soluble starch to improve the utilization efficiency of limited water under drought conditions [45,46].

Under drought conditions, plants adjust stomatal closure, root structure, and hydraulic conductance to reduce growth and improve drought tolerance [42]. The root is very important for water absorption, and good root growth is an important indicator of improved drought tolerance [47]. Expression changes of the plant transcription factor genes can lead to changes in root phenotype. For example, *Arabidopsis CmERF053* transgenic plants have more lateral roots under drought stress [48]. Compared with WT lines, OX lines had significantly increased root surface area, root volume, and absorbing root length but significantly decreased root average diameter and conducting root length (Figure 3). This indicates that ERF194 could weaken the inhibition of root growth under drought stress and promote the growth of fine roots to maintain the water absorption and transportation required for plant growth. Plants should absorb water from deep soils by increasing root length under severe drought conditions. We speculate that this root-structure change of OX enhanced the water-absorption capacity and accelerated the water transport of the root system. This special structure of the root system, in conjunction with the significantly reduced aboveground portion, can enhance the absorption of water while reducing the transpiration rate of water loss. 

When plants are subjected to abiotic stress, a large number of reactive oxygen species are accumulated in the body. Excessive reactive oxygen species and their reaction products can cause oxidative damage to lipids, proteins, carbohydrates, and nucleic acids [49]. SOD, CAT, and POD are the main protective enzymes of the membrane lipid peroxidation defense system, which can protect plants from poisoning by removing excessive reactive oxygen species [32,33,34]. Under CK, the activities of CAT, POD, and SOD in OX plants were slightly lower than those in WT plants. After drought stress, compared with WT, the POD, SOD, and CAT activities of OX lines were higher (Figure 4). Through the GO annotation and GO enrichment analysis, we found that *ERF194* was closely related to antioxidant activity, response to stress, reactive oxygen species metabolic process, response to stimulus, and oxidoreductase activity, acting on peroxide as an acceptor (Figure 5). Under CK, the relative expression levels of nine reactive oxygen species and drought-related genes were lower in OX plants than in WT plants but were significantly induced in OX plants under drought stress (Figure 6). Promoter analysis revealed that these nine genes had *cis*-acting elements that ERF194 can bind Appendix A. ERF194 may play a role in plant stress tolerance through a process of transcriptional regulation [40]. The differences in POD, SOD, and CAT activities between OX plants and the other two lines under CK and DR conditions also confirmed the above results (Figure 4b–d). MDA is one of the membrane lipid peroxidation products and can be used to measure the damage of stress to the cell membrane [32]. The increase of MDA content in the lines with strong drought resistance was smaller than that in the lines with weak drought resistance [32]. Under DR, OX had the lowest MDA content and was significantly different from WT and RNAi. However, under CK, the MDA content of OX plants was slightly higher than that of the other two lines. This indicates that ERF194 reduced the synthesis of MDA in response to drought stress, making the cell membrane of OX more stable under drought stress and more resistant to drought stress. This was consistent with the expression of the above drought-responsive genes, which significantly enhanced the scavenging ability of reactive oxygen species and the anti-oxidation ability of the cell membrane of OX lines (Figure 6).

In addition to the reactive oxygen scavenging system, plants respond to drought stress through an osmoregulatory system. Soluble sugar not only provides energy for plant growth and development but also participates in cell osmotic regulation as a soluble substance. Soluble starch is synthesized by plant photosynthesis, and its content is closely related to the photosynthetic rate. NSC in forest trees have functions such as osmotic regulation, signal transduction, and cavitation repair [37,38]. After drought stress, the soluble sugar, starch, and NSC content of OX were significantly increased. Similar to WT, RNAi did not increase significantly (Figure 4g–i). The significant increase of soluble starch in OX can be converted to more soluble sugars, which can be converted to NSC. Changes in soluble substances in the leaves cause changes in the concentration of cellular osmotic substances, thereby regulating leaf water potential. Plants can regulate water potential to alleviate water deficit under drought conditions [35]. After drought stress, the plant water potential was reduced, but OX had a higher water potential than the other two strains, which made it easy to cope with drought stress (Figure 4e). Leaf relative water content (RWC) is closely related to cell volume and can reflect the balance between water supply and transpiration rate [50] and is also positively correlated with plant stress tolerance [51]. Interestingly, the RWC of OX was the highest in this study, which may explain the higher water potential. These all provide support for the drought tolerance of OX.

## 4. Experimental Materials and Methods

### 4.1. Plant Materials and Drought Treatment

The 30-days-old ERF194-overexpression (OX), -suppression (RNAi) and non-transgenic (WT) hybrid poplar clone 717 subcultured in the medium was cultivated by our group [29]. It was cultured in the greenhouse under conditions of relative humidity 60~70%, photoperiod 16/8 h light/dark cycle, and temperature 25 ± 2 °C of Forestry College of Shanxi Agricultural University. After hydroponics for 20 days, the plants were transplanted to pots with mixed soil (peat: vermiculite: perlite = 7:1.5:1.5). After recovery of growth for one month, 24 plants with uniform growth status were divided into drought treatment and the control groups. The soil water content of the normal watering control group was controlled at 70~80% of the maximum field capacity by the weighing method, and the treatment group was drought-stressed for 18 d after watering well [29]. The growth and development of plants were measured every 9 days.

### 4.2. Characteristics of Leaf

The length (mm), width (mm), area (mm^2^), and perimeter (mm) of the 7th functional leaf on the 18th day of drought stress were measured using a YMJ-C leaf area scanner (YMJ-C, Zhejiang, China) [52]. The PMS1505D-EXP water potential meter (PMS Albany, USA) was used to measure the leaf water potential (Mpa) [53]. Changes in chlorophyll content of fixed leaves were monitored with a handheld chlorophyll meter (CCM-200, USA) [54]. The 7th functional leaf was taken and weighed to obtain its fresh weight (FW). The leaf was transferred to water and soaked for 24 h to obtain turgid weight (TW). Drying the leaves to a constant weight at 65℃ was used to obtain the dry weight (DW). The relative water content (RWC) was calculated as RWC=(FW - DW)/(TW - DW) × 100% [50].

The above-mentioned dried leaves were pulverized, and the soluble sugar and starch contents were determined by the anthrone-sulfuric acid method. Non-structural carbohydrate (NSC) content is designed as the sum of soluble sugar and starch [55]. After 18 days of drought, the fifth functional leaf was taken to determine leaf physiological indices. The nitrogen blue tetrazolium (NBT) photoreduction method was used to measure SOD activity. UV spectrophotometric and guaiacol methods were used to measure CAT and POD activity, respectively. The barbituric acid colorimetric method was used to determine the content of MDA [56]. Leaves were sampled at approximately 10:00 am.

### 4.3. Characteristics of Root

After 18 days of drought, four plants of uniform growth status were selected from each treatment and carefully removed from the pots. The soil and other impurities were washed away with running water, and dead and live roots were distinguished according to the morphological characteristics of the root system. Then, the poplar roots were divided into absorbing roots and conducting roots according to diameters ≤2.0 mm and >2.0 mm [30]. The root-scanning system EPSON EXPRESSION 10,000 XL (resolution set to 400 dpi) was used to scan and photograph, and the average root diameter (mm), surface area (cm^2^), volume (cm^3^), and length (cm) were measured with WinRHIZO 2016p software (Regent Instruments Inc., Canada) [30]. The roots were dried at 80 °C for 48 h, and the dry weight (DW) was weighed. Specific root length (SRL)=Length/DW, specific root area (SRA)=Surf Area/DW, and root tissue density (RTD)=Root Volume/DW were analyzed.

### 4.4. Gene Annotation and Expression Patterns

The 4th leaf and stem internodes isolated from one-month-old WT and three lines of OX grown on 1/2 MS medium were used for RNA-Seq. For OX plants, three independent transgenic lines (materials from at least 12 plants pooled per line) served as three biological replicates. Three samples from the WT (materials pooled from at least 12 plants for each) were harvested and served as three technical repeats. A total of 12 libraries were constructed and sequenced. The sequencing depth is 15× and the statistical power of this experimental design calculated in RNASeqPower is 0.85. Gene expression was reported as transcripts per million reads (TPM). Differentially expressed genes (DEGs) between OX and WT were identified using DESeq2 with a log2fold-change (|log2FC|) ≥1 and an adjusted *p*-value ≤ 0.05 as cutoffs. The construction of sequencing libraries was carried out with reference to previous studies [28,29]. Six genes related to water deficiency (*Potri.001G070900*, *Potri.001G084000*, *Potri.002G013200*, *Potri.008G065600*, *Potri.018G063300*, and *Potri.005G195600*) and three genes related to ROS scavenging (*POD2*, *POD3*, and *SOD2*) were selected as a gene set based on the transcriptome data to detect the co-expression relationship with *ERF194* [29]. The specific gene function was shown in Table 1. The software Goatools and Fisher’s exact test (P_adjust_ < 0.05) were used for gene ontology (GO) enrichment analysis [28].

### 4.5. RNA Extraction and RT-qPCR

On the 9th day after drought treatment, we sampled the 8th functional leaf and immediately froze it in liquid nitrogen for RNA extraction and RT-qPCR analysis [29]. The coding sequence (CDS) region and the amino acid sequence of all genes in the above gene set were downloaded from the *JGI* database (https://phytozome-next.jgi.doe.gov/, accessed on 3 July 2021).The relative expression of the gene was detected using SuperReal PreMix (SYBR Green, TIANGEN, Beijing, China) on Agilent Mx3000 P real-time fluorescence quantitative PCR system with primers as shown in Table 1. The relative expression level was calculated as 2^−ΔΔCt^ [21,29].

### 4.6. Analysis of cis-Acting Elements in the Promoter

The promoter sequences (2kb upstream of the translation start site) were blasted and obtained from the Phytozome database (https://phytozome-next.jgi.doe.gov//, accessed on 28 August 2022) of *ERF194* and nine stress-related genes. The *cis*-elements in promoters were predicted and determined using the PlantCRAE (http://bioinformatics.psb.ugent.be/webtools/plantcare/html/, accessed on 28 August 2022).

### 4.7. Data Processing

SPSS 22.0 software (SPSS Statistics for Windows, Version 22.0. SPSS Inc., Chicago, IL, USA) and one-way ANOVA analysis were used for data processing and variance analysis. Using Duncan’s method, *p*-value < 0.05 was set to test the significance of differences between data. The Origin 2019 b software (OriginLab, Northampton, MA, USA) was used to obtain graphs. Each experiment was repeated at least three times.

## 5. Conclusions

In conclusion, ERF194 reduced water loss by regulating the semi-dwarf morphology and increased fine roots to enhance the ability of absorbing water. It also improved the drought tolerance of poplar by increasing water potential and increasing the relative water content of the leaves. Last but not the least, ERF194 reduced the effects of drought stress on poplar growth by regulating the expression of stress-related genes to increase the activity of antioxidant enzymes and promote the synthesis of soluble sugars and starch as well as to inhibit the products of MDA. All of these stress-related physiology indices were acted out by the resilient growth status and robust root system of OX under drought-stress conditions. These findings will provide new ideas for the physiological functions of the ERF194 transcription factor in plant growth and drought-stress tolerance.

## Figures and Tables

**Figure 1 ijms-24-00788-f001:**
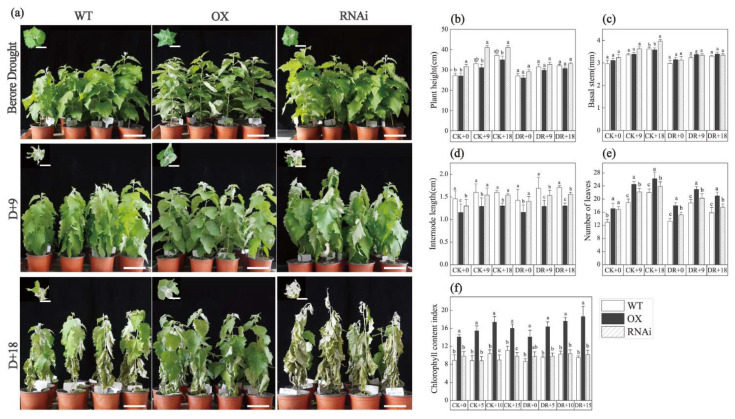
Growth status of transgenic and non-transgenic poplars under drought stress. (**a**) Morphology of WT, OX, and RNAi cuttings cultured for 9 d and 18 d under drought conditions, bars: 10 cm. (**b**) Plant height, (**c**) basal diameter, (**d**) internode length, (**e**) number of leaves, and (**f**) chlorophyll content index. Data represent means ± SD of 12 independent biological samples of transgenic poplar lines (OX or RNAi) and non-transgenic poplars, respectively. Different letters indicate significant dif-ference at *p* < 0.05 level. WT, non-transgenic poplar line; OX, ERF194-overexpressing line; RNAi, ERF194-suppressing line; CK, normal watering conditions; DR, drought-stress conditions.

**Figure 2 ijms-24-00788-f002:**
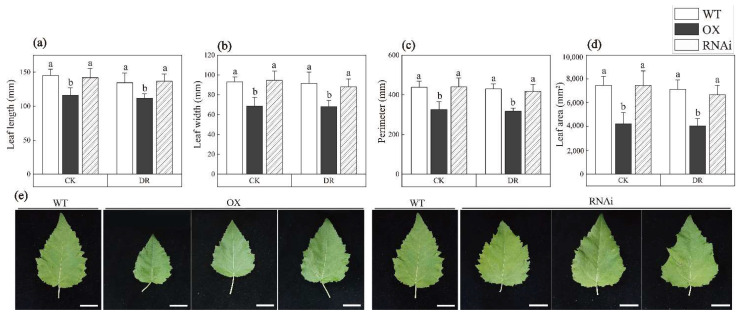
Differences of leaf morphological indices under drought stress. (**a**) Leaf length, (**b**) leaf width, (**c**) perimeter, (**d**) leaf area, and (**e**) leaf morphology of poplar WT, OX, and RNAi seedlings cultured for 18 d under drought conditions. Bars: 3 cm. Data represent means ± SD of 12 independent biological samples of transgenic poplar lines (OX or RNAi) and non-transgenic poplars, respectively. Different letters indicate significant difference at *p* < 0.05 level. WT, non-transgenic poplar line; OX, *ERF194*-overexpressing line; RNAi, *ERF194*-suppressing line; CK, normal watering conditions; DR, drought-stress conditions.

**Figure 3 ijms-24-00788-f003:**
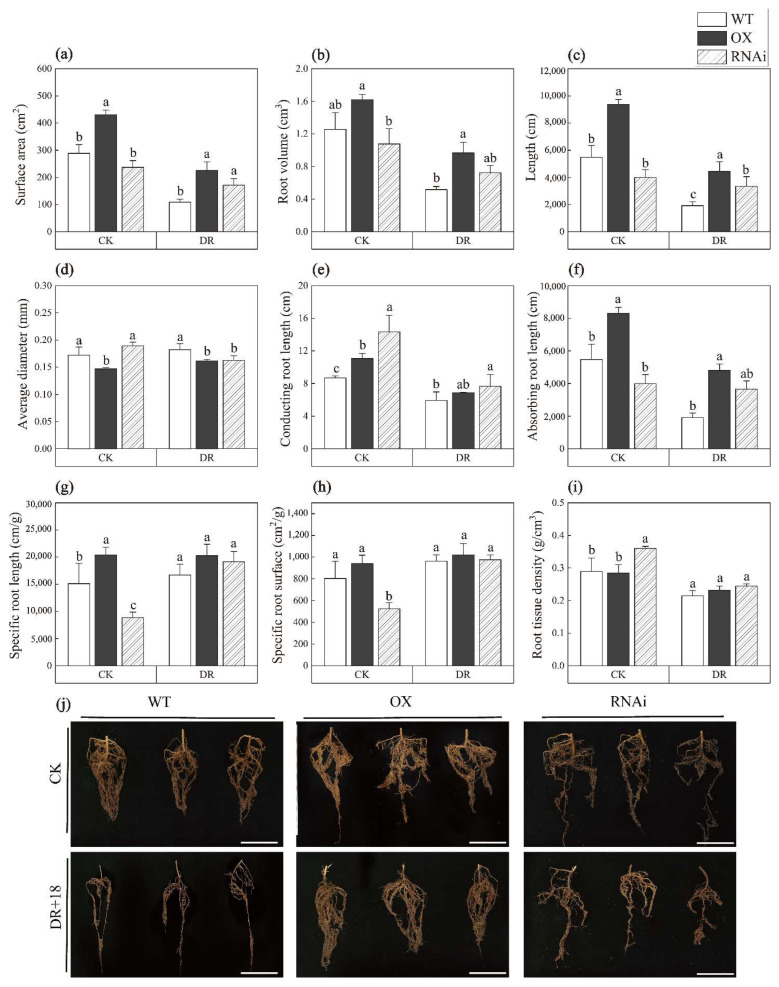
Root differences between transgenic poplar and non-transgenic poplar under drought stress. (**a**) Surface area, (**b**) root volume, (**c**) length, (**d**) average diameter, (**e**) conducting root length, (**f**) absorbing root length, (**g**) specific root length, (**h**) specific root surface, (**i**) root tissue density, and (**j**) root morphology of WT, OX, and RNAi seedlings cultured for 18 d under control and drought conditions. Bars: 8 cm. Data represent means ± SD of 12 independent biological samples of transgenic poplar lines (OX or RNAi) and non-transgenic poplars, respectively. Different letters indicate significant difference at *p* < 0.05 level. WT, non-transgenic poplar line; OX, *ERF194*-overexpressing line; RNAi, *ERF194*-suppressing line; CK, normal watering conditions; DR, drought-stress conditions.

**Figure 4 ijms-24-00788-f004:**
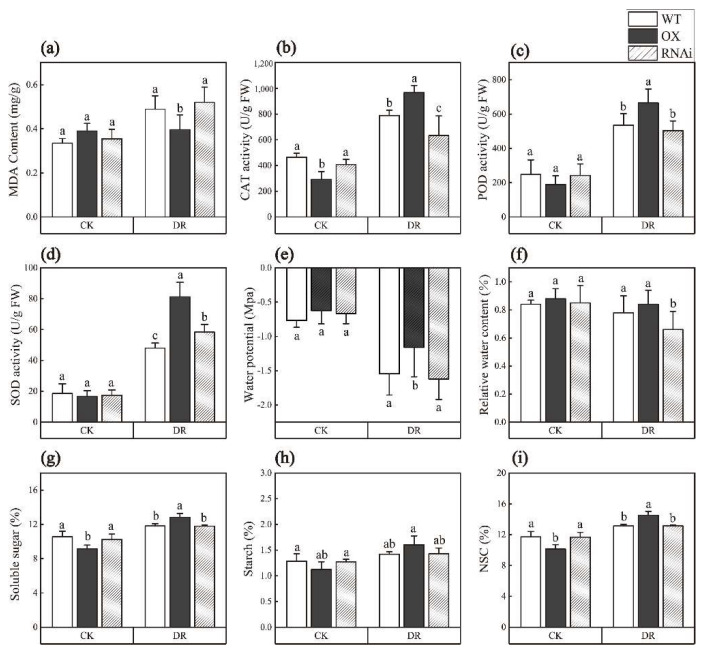
Physiological indices of transgenic and non-transgenic poplars under drought stress. (**a**) MDA, (**b**) CAT, (**c**) POD, (**d**) SOD, (**e**) water potential, (**f**) relative water content, (**g**) soluble sugar, (**h**) starch, and (**i**) NSC. Data represent means ± SD of 12 independent biological samples of transgenic poplar lines (OX or RNAi) and non-transgenic poplars, respectively. Different letters indicate significant difference at *p* < 0.05 level. WT, non-transgenic poplar line; OX, *ERF194*-overexpressing line; RNAi, *ERF194*-suppressing line; CK, normal watering conditions; DR, drought-stress conditions.

**Figure 5 ijms-24-00788-f005:**
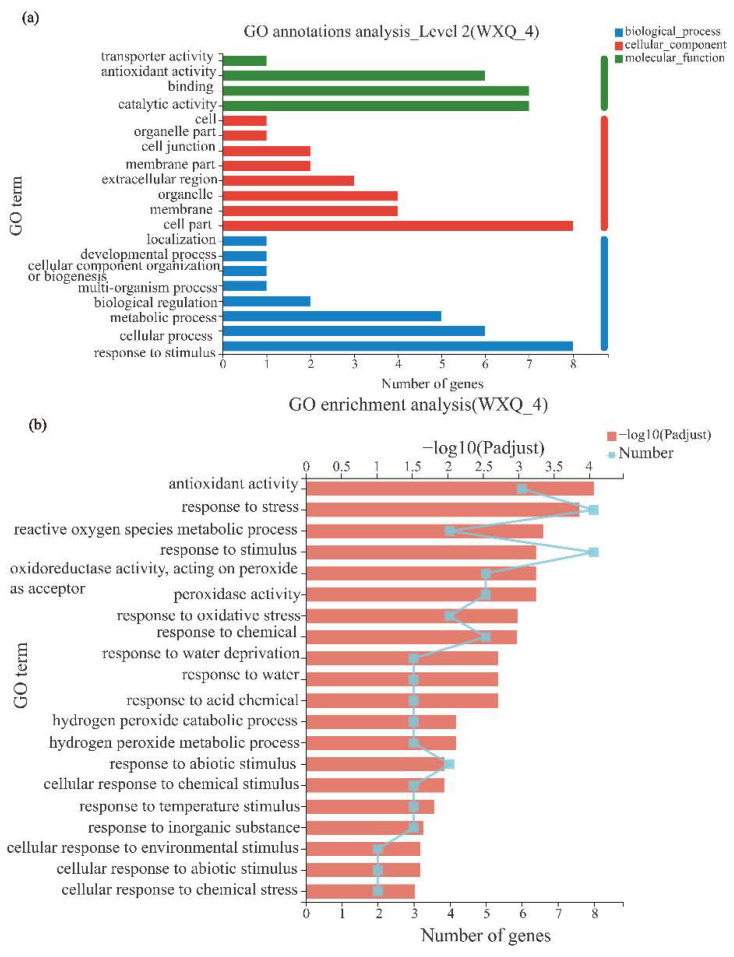
GO annotation and enrichment analysis of gene sets. (**a**) GO annotation analysis; (**b**) GO enrichment analysis.

**Figure 6 ijms-24-00788-f006:**
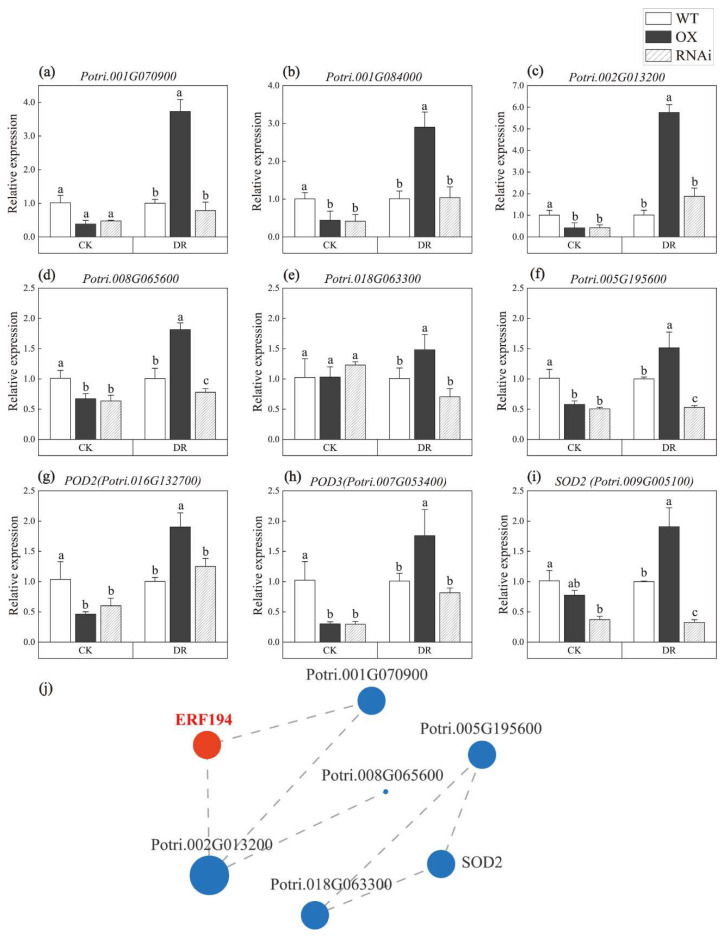
Expression pattern and co-expression analysis of stress related genes. (**a**) *Potri.001G070900*, (**b**) Potri.001G084000, (**c**) Potri.002G013200, (**d**) Potri.008G065600, (**e**) *Potri.018G063300*, (**f**) *Potri.005G195600*, (**g**) *POD2 (Potri.016G132700)*, (**h**) *POD3 (Potri.007G053400)*, (**i**) *SOD2 (Potri.009G005100)*, and (**j**) co-expression analysis. Different letters indicate significant difference at *p* < 0.05 level. WT, non-transgenic poplar line; OX, *ERF194*-overexpressing line; RNAi, *ERF194*-suppressing line; CK, normal watering; DR, drought-treatment conditions.

**Table 1 ijms-24-00788-t001:** Primers used for RT-qPCR and gene function annotation.

Gene_ID	Forward and Reverse Primers (5′-3′)	Gene Function Annotation
*Potri.001G070900*	TCTGAGGACAGTTCGGACACAG GTGAACCGCCACAGAATCGTCAC	Response to oxidative stress; peroxidase activity
*Potri.001G084000*	GGCTGAAACCAAGAACCAACAGG GGCAAGCCTGTAGACAACATTG	Cysteine and methionine metabolism; amino acid transport and metabolism
*Potri.002G013200*	CCAAGACCAGTGCTGGTGAAGAG CAGTCGGAGGAGGAACCTCTTCAG	Response to abscisic acid
*Potri.008G065600*	ATGGAGGGCAAAGAAGAAGATG CCATGAAGTCAGTTCGCCTGGCTC	Mediate rapid entry or exit of water in response to abrupt changes in osmolarity; carbohydrate transport and metabolism
*Potri.018G063300*	CTCTGCCTAACCACTCTCTACC CTTGAGGCCATAGAACTGTGACCG	Response to oxidative stress; peroxidase activity
*Potri.005G195600*	GGCCAGAGCTGCTATGTCCTTTAC CCATTCTCTATAGGGATGGTGCC	Response to oxidative stress; peroxidase activity
*POD2 (Potri.016G132700)*	CAGGCTGCTTTCAGGACAGACTTTG GATCTGGCCATTTGTGCCAGTAAGTG	Response to oxidative stress; peroxidase activity; oxidation-reduction process
*POD3 (Potri.007G053400)*	GACTCCAGAATAGCCATCAACATGG GGCTTGTTGAAAGGCCTGTGAGTC	Response to oxidative stress; peroxidase activity
*SOD2 (Potri.009G005100)*	CTAATGTTGAAGGCGTCGTC ACGCATCCATTTGTTGTGTC	Response to oxidative stress; superoxide dismutase activity
*ERF194*	ATGGTGAGAGAAAGAAGGGAGC TCATTAAACTGTCCACACACCAGG	
*Actin*	ACCCTCCAATCCAGACACTG TTGCTGACCGTATGAGCAAG	

## Data Availability

The datasets generated during and/or analyzed during the current study are available in online repositories. The names of the repository/repositories and accession number(s) can be found at: https://submit.ncbi.nlm.nih.gov/, PRJNA760939, accessed on 1 June 2021.

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
