# Peer review of "Transcription Factor ERF194 Modulates the Stress-Related Physiology to Enhance Drought Tolerance of Poplar"

_ijms, 2023, doi:10.3390/ijms24010788_

Round 1

Reviewer 1 Report

This paper investigates the role of the ERF194 transcription factor on the growth and drought tolerance of poplar.  The authors used a glasshouse drought experiment to compare morphological, physiological and gene expression in young poplar plants where ERF194 was overexpressed (OX), supressed (RNAi) and non-transgenic poplar (WT).  The authors have found that OX increases plant drought tolerance by increasing fine root production and decreasing above ground biomass. In addition OX plants had higher water potential, clorophyll content and NSC concentration. 

The paper results are not groundbreaking but provide some useful information about the role of the ERF194. The paper experimental design and methodology seems appropriated, even though more information is needed about some measurements (i.e. water potential) and statistical analyses. 

The text is very poorly written and needs to be extensively revised before it can be published. There are also some conceptual problems about the interpretation of the physiological data and plant physiology in general. I'm not sure if these problems are just poorly worded sentences or they are really conceptual misunderstandings about plant physiology. Regardless they need to be corrected. See my specific comments below:

L46: Needs to include hydraulic damage to plant hydraulic system to the list of drought effects. 

L46: Probably should use "reduced" instead of "weakened"

L53: If the photosynthesis was reduced it means the plant did not resist the drought

L53: "Current climate model" makes no sense. There isn't a single "current" climate model. 

L93: What "Protective forest species" means?

L128: How many leaves in how many plants? at what time of the day? 

L192: Have the anova assumptions (residual normality and homoscedasticity) been tested? How?

L194: What does that means? Does the presented results includes data from all experiments or just some of them? 

L197-198: This first section should not be in the results section.

L238-239: Needs to add reference.

L283: Water potential by itself does not reflect drought resistance.  Even very drought vulnerable plants can have very negative water potential if they are exposed to severe water stress. The difference is that drought resistant plants can survive this low water potential whereas vulnerable plants cannot. 

L292: Non-structural carbohydrates

L355: What "by relating expressed genes" means?

L357: "and enhance drought tolerance to adaptation" This doesn't mean anything. 

L359: There isn't a single most important organ for plant growth. If there aren't enough roots or transporting tissue in the stem, the plant wont growth regardless of its leaf area. 

L361: "Outside world" its a weird choice of words. I suggest using "environment" instead. 

L365: "have the characteristics of plant height" This doesn't mean anything. 

L370: Replace "always" with "often". Evergreen plants can exist in dry ecosystems. 

L386: "Absorb" instead of "suck up"

L426-427: You're repeating the text from the results. Also see the comments on the results regarding the relationship between water potential and drought resistance. Water potential by itself does not indicate plant drought resistance. 

L439: OX leaves had the highest water potential, that is, less negative values. 

L442: That explains the higher water potential. Water potential is related to the tissue water content (see a pressure volume curve). 

L447: You have no direct measurements of water loss, so you should not conclude anything about it. Considering OX have more absorbing roots, its possible that ERF194 have a role increasing water absorption instead of reducing water loss. 

Author Response

Response to reviewer

 Reviewer 1

This paper investigates the role of the ERF194 transcription factor on the growth and drought tolerance of poplar.  The authors used a glasshouse drought experiment to compare morphological, physiological and gene expression in young poplar plants where ERF194 was overexpressed (OX), supressed (RNAi) and non-transgenic poplar (WT).  The authors have found that OX increases plant drought tolerance by increasing fine root production and decreasing above ground biomass. In addition OX plants had higher water potential, clorophyll content and NSC concentration. 

The paper results are not groundbreaking but provide some useful information about the role of the ERF194. The paper experimental design and methodology seems appropriated, even though more information is needed about some measurements (i.e. water potential) and statistical analyses. 

The text is very poorly written and needs to be extensively revised before it can be published. There are also some conceptual problems about the interpretation of the physiological data and plant physiology in general. I'm not sure if these problems are just poorly worded sentences or they are really conceptual misunderstandings about plant physiology. Regardless they need to be corrected. See my specific comments below:

A: We have made a comprehensive and systematic revision based on the reviewers' comments and suggestions.Please check, thank you.

L46: Needs to include hydraulic damage to plant hydraulic system to the list of drought effects. 

A: We have corrected it, thank you.

L46: Probably should use "reduced" instead of "weakened"

A: We have corrected it, thank you.

L53: If the photosynthesis was reduced it means the plant did not resist the drought

A: Thank you for pointing out the error, we have fixed the clerical error, thank you.

L53: "Current climate model" makes no sense. There isn't a single "current" climate model. 

A: Thank you for pointing out, we have revised, please check, thank you.

L93: What "Protective forest species" means?

A: It menas “shelter forest”. We have revised, please check, thank you.

L128: How many leaves in how many plants? at what time of the day? 

A:There were 12 plants in each treatment. The seventh functional leaf of each plant was used to measure water potential, relative leaf water content, leaf length, leaf width and leaf area, and nonstructural carbohydrates. The fifth functional leaf was used to measure antioxidant enzymes and malondialdehyde activity. Leaves were sampled at approximately 10:00 am.

L192: Have the anova assumptions (residual normality and homoscedasticity) been tested? How?

A:The data were analyzed by ANOVA. Details can be found in the Supplementary file.

L194: What does that means? Does the presented results includes data from all experiments or just some of them? 

A:One-way ANOVA was used, using Duncan's method with a P value < 0.05 test. All data were analyzed in this way.

L197-198: This first section should not be in the results section.

A:Thank you for your suggestion, we have made the changes. Please check, thank you.

L238-239: Needs to add reference.

A:Thank you for your suggestion, we have added the reference, please check it, thanks.

L283: Water potential by itself does not reflect drought resistance.  Even very drought vulnerable plants can have very negative water potential if they are exposed to severe water stress. The difference is that drought resistant plants can survive this low water potential whereas vulnerable plants cannot. 

A:Thank you for your corrections, we have made changes to the relevant parts. Please check. Thank you.

L292: Non-structural carbohydrates

A:Thank you for pointing out the error, it has been corrected, please check. Thank you.

L355: What "by relating expressed genes" means?

A: We have revised it.The morphology and physiological metabolism of plants are the results of gene expression and environmental factors. please check, thank you.

L357: "and enhance drought tolerance to adaptation" This doesn't mean anything. 

A:Thank you for your suggestion, we have made the changes. Please check, thank you.

L359: There isn't a single most important organ for plant growth. If there aren't enough roots or transporting tissue in the stem, the plant wont growth regardless of its leaf area. 

A:Thank you for your comments, this statement is indeed not quite right, we have made changes. Please check, thank you.

L361: "Outside world" its a weird choice of words. I suggest using "environment" instead. 

A:Thank you for your suggestion, we have made the changes. Please check, thank you.

L365: "have the characteristics of plant height" This doesn't mean anything. 

A:Thank you for your suggestion, we have made the changes. Please check, thank you.

L370: Replace "always" with "often". Evergreen plants can exist in dry ecosystems. 

A:Thank you for your suggestion, we have made the changes. Please check, thank you.

L386: "Absorb" instead of "suck up"

A:Thank you for your suggestion, we have made the changes. Please check, thank you.

L426-427: You're repeating the text from the results. Also see the comments on the results regarding the relationship between water potential and drought resistance. Water potential by itself does not indicate plant drought resistance. 

A:Thank you for your suggestion, we have reopened the discussion. Please check, thank you.

L439: OX leaves had the highest water potential, that is, less negative values. 

A:Thank you for your suggestion, we have reopened the discussion. Please check, thank you.

L442: That explains the higher water potential. Water potential is related to the tissue water content (see a pressure volume curve). 

A:Thank you for pointing this out, we have made the changes.Please check, thank you.

L447: You have no direct measurements of water loss, so you should not conclude anything about it. Considering OX have more absorbing roots, its possible that ERF194 have a role increasing water absorption instead of reducing water loss. 

A:Thank you for pointing this out, we have made the changes.Please check, thank you.

Reviewer 2 Report

Make fonts uniform. The abstract has larger fonts than the rest of the text.

Address minor English grammar errors.

Improve Fig-5 for font size, uniformity, resolution, and visibility.

Author Response

Response to reviewer

 Reviewer 2

Comments and Suggestions for Authors

Make fonts uniform. The abstract has larger fonts than the rest of the text.

A:Thank you for pointing this out, we have made the changes.Please check, thank you.

Address minor English grammar errors.

A:Thank you for pointing this out, we have made the changes.Please check, thank you.

Improve Fig-5 for font size, uniformity, resolution, and visibility.

A:Thank you for pointing out the problem, we have revised the font and image, and further improved the syntax.Please check, thank you.

Reviewer 3 Report

The manuscript entitled “Transcription factor ERF194 modulates the stress-related physiology to enhance drought tolerance of poplar”  intended for  publication in International Journal of Molecular Sciences is an interesting paper, however I think that manuscript still needs some improvements.

I have mainly minor remarks. Generally, the paper is relatively straightforward and well written, however some parts of manuscript need more attention. The Authors should improve Keywords (remove some title repetitions) and more specify/improve the purpose of the study. I think, the Authors could modify the beginning of Introduction and Discussion. Authors should check more carefully the Reference list, and improve it.  In addition, there are small mistakes in the text of manuscript, including Reference list, that need to be corrected by Authors (e.g. lines: 490, 495, 510, 520, 606, 653, 655, 675, 677, 694, 704, 708, 712, 716, 748, 767, 786).

Author Response

Response to reviewer

 Reviewer 3

Comments and Suggestions for Authors

The manuscript entitled “Transcription factor ERF194 modulates the stress-related physiology to enhance drought tolerance of poplar”  intended for  publication in International Journal of Molecular Sciences is an interesting paper, however I think that manuscript still needs some improvements.

I have mainly minor remarks. Generally, the paper is relatively straightforward and well written, however some parts of manuscript need more attention. The Authors should improve Keywords (remove some title repetitions) and more specify/improve the purpose of the study. I think, the Authors could modify the beginning of Introduction and Discussion. Authors should check more carefully the Reference list, and improve it.  In addition, there are small mistakes in the text of manuscript, including Reference list, that need to be corrected by Authors (e.g. lines: 490, 495, 510, 520, 606, 653, 655, 675, 677, 694, 704, 708, 712, 716, 748, 767, 786).

A: Thanks to the suggestions you gave, we have improved the keywords and the purpose of the study. Also revised the introduction and the beginning of the discussion. Finally the reference list was checked and updated. Please check it, thank you.
